# Phylogenetic History and Phylogeographic Patterns of the European Wildcat (*Felis silvestris*) Populations

**DOI:** 10.3390/ani13050953

**Published:** 2023-03-06

**Authors:** Edoardo Velli, Romolo Caniglia, Federica Mattucci

**Affiliations:** Unit for Conservation Genetics (BIO-CGE), Italian Institute for Environmental Protection and Research (ISPRA), Via Cà Fornacetta 9, 40064 Ozzano dell’Emilia, Italy

**Keywords:** European wildcat, *Felis silvestris*, divergence times, glacial *refugia*, mitochondrial DNA, mito-nuclear discordances, phylogeny, phylogeography

## Abstract

**Simple Summary:**

The European wildcat is an iconic small-sized predator that is still threatened by habitat fragmentation, accidental or illegal killings, and hybridization with domestic cats. However, phylogenetic and phylogeographic patterns of the *taxon*, though essential to design appropriate long-term conservation management actions, have been poorly investigated. Therefore, in this study, for the first time, we describe the most geographically-representative evolutionary history of the species in Europe based on mitochondrial DNA sequences. Our results clearly show that the European wildcat genetic variability was mainly originated during Pleistocene climatic oscillations and successively modeled by both historical natural gene flow among wild lineages and more recent wild *x* domestic anthropogenic hybridization events.

**Abstract:**

Disentangling phylogenetic and phylogeographic patterns is fundamental to reconstruct the evolutionary histories of *taxa* and assess their actual conservation status. Therefore, in this study, for the first time, the most exhaustive biogeographic history of European wildcat (*Felis silvestris*) populations was reconstructed by typing 430 European wildcats, 213 domestic cats, and 72 putative admixed individuals, collected across the entire species’ distribution range, at a highly diagnostic portion of the mitochondrial ND5 gene. Phylogenetic and phylogeographic analyses identified two main ND5 lineages (D and W) roughly associated with domestic and wild polymorphisms. Lineage D included all domestic cats, 83.3% of putative admixed individuals, and also 41.4% of wildcats; these latter mostly showed haplotypes belonging to sub-clade Ia, that diverged about 37,700 years ago, long pre-dating any evidence for cat domestication. Lineage W included all the remaining wildcats and putative admixed individuals, spatially clustered into four main geographic groups, which started to diverge about 64,200 years ago, corresponding to (i) the isolated Scottish population, (ii) the Iberian population, (iii) a South-Eastern European cluster, and (iv) a Central European cluster. Our results suggest that the last Pleistocene glacial isolation and subsequent re-expansion from Mediterranean and extra-Mediterranean glacial *refugia* were pivotal drivers in shaping the extant European wildcat phylogenetic and phylogeographic patterns, which were further modeled by both historical natural gene flow among wild lineages and more recent wild *x* domestic anthropogenic hybridization, as confirmed by the finding of *F. catus*/*lybica* shared haplotypes. The reconstructed evolutionary histories and the wild ancestry contents detected in this study could be used to identify adequate Conservation Units within European wildcat populations and help to design appropriate long-term management actions.

## 1. Introduction

Biogeographic and phylogeographic patterns are key factors in clarifying the evolutionary histories of *taxa* and identifying appropriate conservation management units (ESU and MU [1,2]) to be long-term preserved [3,4]. Quaternary climate oscillations have significantly affected the current composition, distribution, and genetic diversity of species, subspecies, and populations, with local extinctions, replacements, and re-expansions with secondary contacts within the Palaearctic region [5]. Additionally, the current Eurasian *taxa* distribution and genetic structure have been further altered by more recent anthropic factors, such as natural habitat fragmentation, domestication, and alien species invasions, which are heavily impacting biodiversity and ecosystem equilibria [6,7]. The potential consequences of such past environmental changes and recent intense anthropic activities on the survival of natural *taxa* have been extensively investigated through a number of molecular studies, which allowed to reconstruct complex phylogeographic structures [8,9,10], identify local adaptations [11,12], trace migration routes [13,14], detect domestic or allochthonous introgression signals [15,16], and design sound conservation strategies, especially for populations affected by protracted demographic declines [17,18].

Among Eurasian felids, the European wildcat (*Felis silvestris*), heterogeneously distributed throughout Europe, Caucasus, and Turkey, represents a challenging conservation priority, potentially threatened by habitat loss and anthropogenic hybridization, whose taxonomy and systematics are, however, still debated [19,20,21].

Indeed, based on genetic and biogeographic information, Driscoll et al. [20] included the European wildcat among the five subspecies of *Felis silvestris* (*F. s. silvestris*, *F. s*. *lybica*, *F. s*. *catus*, *F. s. ornata*, and *F. s*. *cafra*). Conversely, Kitchener et al. [22] recently reclassified the European *taxa* as three separate species: the European wildcat, *Felis silvestris*, the African wildcat, *Felis lybica*, and the domestic cat, *Felis catus*.

Fossil records and archaeological remains dated the first appearance of the European wildcat in the continent back to 450,000–200,000 years ago [23]. Since then, both Pleistocene natural climatic oscillations [24,25] and anthropogenic events, such as human persecution [26], deforestation, habitat modifications, and local decline of major prey [27,28,29], have continuously shaped current distribution patterns and genetic structure of the species [25]. In particular, the European wildcat genetic structure has been further muddled by anthropogenic hybridization and subsequent introgression with domestic cats, which were human-spread from the Middle East through Europe after their domestication from the African wildcat [20,30,31].

To shed light on the conservation status of the *taxon*, the European wildcat population genetic structure [32,33,34] and patterns of domestic admixture [35,36,37,38,39] have been deeply addressed, principally using nuclear molecular (STR and SNP) markers, detecting five main biogeographic groups and variable degrees of domestic introgression throughout Europe [25,34,40].

However, only a few molecular studies have investigated phylogenetic and phylogeographic patterns of *Felis silvestris* in Europe so far, mainly using informative mitochondrial portions (such as the Control Region or the NADH Dehydrogenate Subunits) to distinguish domestic cat from European wildcat lineages, detecting the rough presence of a continental and Mediterranean clade among the latter [20,31,35,36,41,42]. However, these studies also hightailed the occurrence of anomalous cases of *F. catus*/*lybica* mitochondrial lineages among nuclear-pure wildcat individuals, which might be interpreted as (i) the legacy of ancient natural [31], (ii) Neolithic human-mediated [41] hybridization events between European and African wildcat populations, or (iii) a signal of a more recent anthropogenic admixture between European wildcats and domestic cats [20]. Nevertheless, only a small number of individuals were typed in these studies, and the analyzed samples were collected across restricted geographical areas. Thus, an extensive and exhaustive analysis of the European wildcat mitochondrial diversity patterns across the continent is still missing and would be necessary to disentangle the complex evolutionary framework of the species and better understand the role played by natural and anthropogenic drivers.

In this study, therefore, we analyzed variation patterns of a portion of the mitochondrial NADH dehydrogenase subunit 5 in a geographically representative set of European wildcat and domestic cat individuals with the scope to (1) accurately describe European wildcat phylogenetic and phylogeographic patterns in Europe, (2) date back divergence times among the main haplogroups in the European wildcat phylogenetic history, and (3) evaluate different likely biogeographic scenarios to clarify the origin of shared haplotypes between wild and domestic cats.

## 2. Materials and Methods

### 2.1. Sampling

A total of 705 high-quality DNA cat samples, opportunistically collected from found dead animals between 1998 and 2010 in 12 sub-regions belonging to six European biogeographic macro-regions (Figure 1), were selected from the ISPRA *Felis* DNA biobank [25]. Sampled individuals had been previously identified morphologically by collectors according to phenotype, life history traits, and biometric indices [43,44,45]. Samples had also been previously genotyped at 31 microsatellites (STR) *loci* [25] and genetically classified as domestic (*Felis catus*), wild (*Felis silvestris*), or putative wild *x* domestic admixed cats through Bayesian clustering analyses based on an assignment threshold of posterior probability proportion to belong to the wildcat cluster *q*_i_ = 0.90 [33] (Appendix A). According to these criteria, the selected samples were re-classified into 420 European wildcats, 213 domestic cats, and 72 putative admixed individuals (see Mattucci et al. [25] for details about sampling, DNA extraction, genotyping, and assignment methods). Due to the high level of domestic introgression spread in Scotland and Hungary, felid samples from these two sub-regions were mainly represented by wild *x* domestic admixed cats [25].

### 2.2. Mitochondrial DNA Sequencing and Haplotype Identification

Selected samples were sequenced for a portion of 835 bp of the mitochondrial NADH dehydrogenase subunit 5 (ND5; nucleotides 13,149–13,983 mapped on the mitochondrial genome of the domestic cat; NCBI Reference Sequence NC001700). This mtDNA region is particularly suitable for phylogeny reconstructions, because of the absence of nuclear mitochondrial segments (numts) and its reduced homoplasy, and is highly discriminating, since it contains seven diagnostic mutations useful to distinguish the European wildcat (*F. silvestris*) from the domestic/African cat (*F. catus*/*lybica*) lineages [20,31].

Fragments were amplified using Polymerase Chain Reaction (PCR) primers F2B (5′-TGCCGCCCTACAAGCAAT-3′) and R3B (5′-TAAGAGACGTTTAATGGAGTTGAT-3′) [47]. Each 10 µL PCR reaction contained 2 µL of DNA (c. 50 ng), 0.8 µL of 10X Taq Buffer advanced (Eppendorf) with self-adjusting Mg²^+^ (Eppendorf), 0.80 µL of 0.2% bovine serum albumin (Sigma–Aldrich, St. Louis, MO, USA), 0.36 µL of 2.5 mM dNTPs (Eppendorf), 0.15 µL of each 10 mM primer solution (Bionordika, Solna, Sweden), 0.04 µL of 5 U/µL HotStart Taq polymerase (Eppendorf, Tokyo, Japan), and 5.70 µL of purified water (Eppendorf, Milano, Italy). PCRs were performed in a Veriti Thermal Cycler (Life Technologies, Carlsbad, CA, USA) with the following thermal profile: 94 °C for 15 min for initial denaturation and Taq activation, followed by 50 cycles of 30 s at 94 °C, 60 s at 55 °C and 60 s at 72 °C. The PCR cycling was followed by a final extension for 10 min at 72 °C. PCR products were stored at 4 °C and then purified by exonuclease digestions (1 µL of EXO-SAP per sample, incubated at 37 °C for 30 min, then at 80 °C for 15 min). The purified amplicons were Sanger-sequenced in both directions. Each 10 µL reaction contained 1 µL of amplified DNA, 1 µL of BigDye v1.1 (Life Technologies), 0.2 µL of either the forward or reverse primer, and 7.8 µL of purified water. Sequencing was performed in a Veriti Thermal Cycler with 25 cycles of 10 s at 96 °C, 5 s at 55 °C, 4 min at 60 °C and storage at 4 °C. The purified products were added with 10 µL of Hi-DI formamide (Life Technologies), denatured for 3 min at 95 °C, and analyzed on an Applied Biosystems (ABI) 3130 XL DNA Analyzer. 

### 2.3. Phylogenetic Analyses and Estimates of Divergence Times

The 705 cat NAD5 sequences were aligned and corrected in SeqScape v2.5 (Life Technologies) and used to build a cat NAD5 dataset, which also included other 10 different European wildcat homologous sequences (accession numbers: EF587158, EF587164, EF587168, EF587166, EF587170, EF587169, EF587171, EF587156, EF587162, EF587159; [20]), retrieved from GenBank. 

All the 715 produced and downloaded cat sequences were consequently trimmed into equal sequences of 669 bp (positions 13,243–13,911) to maintain full-length, double-stranded, high-quality sequence data across all samples, using BioEdit v7.1.11 [48].

Identical haplotypes were identified and collapsed using DnaSP v5.10.01 [49], and possible correspondences with haplotypes already published in GenBank were checked using Blast [50]. The final dataset of unique cat NAD5 haplotypes was then used to perform all the downstream diversity, phylogeographic and phylogenetic analyses. 

Haplotype (Hd) and nucleotide diversity (π) were computed using DnaSP. 

The best nucleotide substitution model scheme was computed in PartitionFinder v2 [51] by the Bayesian information criterion (BIC), and, subsequently, three phylogenetic trees were constructed through different computational approaches: (i) the neighbor-joining (NJ, [52]) method, using Smart Model Selection (SMS, [53]) algorithms implemented in Mega v11.0 [54] and performing 10,000 random bootstrap replications; (ii) the maximum-likelihood method (ML, [55]) implemented in Phyml v3.0 [56] with the heuristic search by topological rearrangement of an initial tree (Near-Neighbor-Interchange) and 5000 random bootstrap replications; (iii) the Bayesian method (BT) implemented in Beast v2.1.3 [57], which further allowed to estimate divergence times among nodes. Due to the strong relationship between *taxa*, a strict molecular clock model with a fixed mean substitution rate (2.28 × 10^−8^/site/year [47]) and constant population size as coalescent priors were selected. The Bayesian posterior probabilities (BPPs), as well as the high posterior densities for the node ages (HPDs), were extrapolated by performing three independent MCMC runs of 100,000,000 steps with a burn-in period of 10,000,000 steps and picking genealogies every 2000 steps. The results of the three chains were simultaneously analyzed in Tracer v1.7 [58].

The corresponding portion of the ND5 sequence of *Felis* margarita retrieved from GeneBank (Accession number: EF587034) was used as an outgroup in NJ and ML tree reconstructions. Conversely, the BT tree was calculated without the inclusion of an outgroup, as suggested by Beast developers [57], and the tree rooting point was estimated using as a prior calibration point the time interval in which the common ancestor between *F. silvestris* and *F. catus*/*lybica* likely coalesced (230,000–173,000 years ago [20]).

Each supported node was annotated with bootstrap values for NJ and ML trees and the highest posterior density (HPD) for the BT tree (e.g.: NJ/ML/HPD).

### 2.4. Demographic Analyses

A median-joining (MJ) network analysis was performed in Network v4.6 (Fluxus Technology Ltd., Stanway, UK [59]) to corroborate tree reconstructions, and investigate haplotypes relationships, frequencies, and geographic repartitions, using an ε = 10 and transversions/transition weighting of 3:1.

Haplotype pairwise genetic distances among cat *taxa* and cat biogeographic clusters, assessed with an analysis of molecular variance (AMOVA), and the ϕ_ST_ and ϕ_SC_ indexes for genetic differentiation [60], were computed in Arlequin v3.5.1.3 [61], running 10,000 permutations to evaluate the significance of each parameter.

Further spatial analysis of molecular variance (SAMOVA) was performed using Samova v2.0 [62], which defines clusters of geographically homogeneous populations, based on an a priori definition of the number of *K* groups, and uses a simulated annealing procedure to maximize the proportion of total genetic variance between groups with an AMOVA approach. We tested a different number of groups (with *K* from 2 to 9), each time with the simulated annealing process repeated 10,000 times, starting with a different partition of the population samples into the *K* groups. The selection of the best *K*-repartitions was based on the highest significant values of the F_CT_ genetic differentiation index. F_CT_ estimates differentiation among those groups of populations. The closer F_CT_ is to 1, the more divergent the groups are from each other.

### 2.5. Approximate Bayesian Computation Analyses

The Approximate Bayesian Computation (ABC) simulations [63], implemented in Diyabc v2.1.0 [64], were run to model plausible demographic scenarios and estimate divergence times (in generations). We performed two types of ABC analyses: (a) considering only populations carrying wildcat mitochondrial haplotypes [20]; and (b) considering populations showing mito-nuclear discordance (see Results). In details, to avoid over-computation, we designed the smallest number of evolutionary scenarios using as prior populations the spatially geographical and genetically homogeneous clusters found through SAMOVA analyses. Successively, we simulated alternative evolutionary hypotheses using haplotype distribution and divergence times estimated from the reconstructed Bayesian phylogeny (see Results) and modeled population dynamics, taking into account the main phylogeographic findings reported in the literature [20,25,31]. Therefore, we tested four demographic scenarios for the wildcat haplotypes (Appendix A), hypothesizing that the four clusters split sequentially (scenarios 1 and 3) or simultaneously (scenarios 2 and 4) and that Cluster 4 diverged by isolation (scenarios 1 and 2) or followed a gene flow with other European populations (scenarios 3 and 4). The haplogroup, including wildcats showing mito-nuclear discordances, was analyzed by testing three different scenarios: (a) the three clusters split in recent times from few common ancestral haplotypes (scenario 1); (b) the three clusters split in different sequential evolutionary events (scenario 2); (c) same as scenario 2 but considering longer coalescence time (scenario 3). We ran 4 × 10^6^ simulations for each scenario using prior uniform distributions of the effective population sizes and time parameters with the gamma-distributed mutation model with Gamma shape = 4.0. Scenarios were compared by estimating posterior probabilities using the logistic regression method using 1% of the simulated datasets. For the best models, posterior distributions of the parameters were estimated with a logit-transformed linear regression on the 1% simulated datasets closest to the observed data. Scenario confidence was evaluated by comparing observed and simulated summary statistics. Finally, the goodness-of-fit of the posterior parameters for the best-performing scenarios was tested via the model checking option with default settings, and significance was assessed after Bonferroni correction for multiple testing.

## 3. Results

Haplotype alignment did not show any indels or stop codons and the aminoacidic sequence was concordant with the domestic cat ND5 protein sequence (NCBI Reference Sequence NC001700). Thus, we excluded the amplification of numts or pseudogenes. After regrouping procedures, we identified 29 haplotypes among the 715 sequences, counting 32 polymorphic sites, including 23 parsimony informative sites, with a total haplotype diversity Hd = 0.862 ± 0.006 and a nucleotide diversity π = 0.870 ± 0.013 (Table 1). We compared the resulting haplotype sequences with the NCBI nucleotide database using the Blast algorithm founding 12 new unpublished haplotypes (Appendix A).

### 3.1. Phylogenetic Analyses and Divergence Time Estimates

The best fit evolutionary model for the 29-haplotype alignment was Kimura’s two-parameter (K80 [65]) model with invariable sites (*I* = 0.80) for the first and second codon positions, whereas the Hasegawa, Kishino, and Yano (HKY [66]) model with a gamma distribution and four discrete categories was selected for the third codon position. 

The NJ, ML, and BT phylogenetic trees showed very concordant topologies for the main clades (Appendix A); thus, we described in detail directly the topology of the tree generated by Beast, which presented posterior probabilities of the main internodes *>* 0.90 (Figure 2a). In the BT tree, haplotypes were clearly split into two main and strongly supported lineages (node 1, 100/100/1), diverging 197,500 years ago (95% HPD 173,002–226,274 years ago): (i) lineage D, including 15 haplotypes, shared in 448 (62.3%) individuals and (ii) lineage W, including 14 haplotypes, shared in 267 individuals (37.7%) (Table 1). Haplotypes were separated into three main categories according to the previous 31-STR Bayesian assignment tests performed by Mattucci et al. [25]: (i) category *“d”* included haplotypes found only among domestic cats; (ii) category “*dw*” included haplotypes found either in domestic, wild or putative admixed individuals; and iii) category *“w”* included haplotypes found only among wildcats and putative admixed individuals.

Lineage D included eight haplotypes, identified by the prefix “*d”*, exclusively detected in 25 domestic cat genotypes, and other seven haplotypes, identified by the prefix “*dw*”, found in 174 wildcat (41.4% of the wild), 188 domestic cats, and 61 putative admixed (84.7% of the wild) genotypes (Appendix A). In this lineage, we found the first highly supported split (node 2, 100/88/0.99) dating back about 80,000 years ago (95% HPD 31,561–145,850 years ago), distinguishing two major clade groups, I-II and III (Figure 2a). A second split occurred at node 3 (63/65/0.54), about 50,000 years ago (95% HPD 16,756–97,189 years ago), separating clades I and II (Figure 2a). A final third split appeared at node 4 (61/64/0.92), dating back 37,700 years ago (95% HPD 11,992–76,931 years ago) and dividing sub-clades Ia and Ib (Figure 2a). Interestingly, most wild individuals showing discordant mtDNA variants (130 wildcats) shared *dw*4 and *dw*6 haplotypes within sub-clade Ia, and all 61 putative admixed individuals carried *dw* haplotypes. 

Lineage W included 14 haplotypes, identified by the prefix “*w*”, and shared by 246 (58.6%) wildcats, 11 (15.3%) putative admixed individuals, and all the 10 GeneBank European wildcat reference sequences. It presented only one main supported diverging point (node 5, 96/100/1), dated back to 62,400 years ago (95% HPD 21,860–118,910 years ago), separating two clades, IV and V (Figure 2a) (see Supplementary Material and Appendix A for details).

### 3.2. Phylogeographic Analyses of European Wildcat and Putative Admixed Individuals

MJ network, reconstructed using the entire 29-haplotype alignment, was highly concordant with the topology of the BT tree (Figure 2), identifying two main haplogroups (D and W), which were clearly separated by seven diagnostic polymorphisms, previously described by Driscoll et al. [47], though one of them (D) included seven domestic–wild shared haplotypes (Figure 2b). The AMOVA (Table 2), performed considering European wildcats, domestic cats, and putative admixed cats as different groups, detected a higher proportion of variation within (63.3%) than among (36.7%) populations and a significant differentiation index ϕst = 0.37 (*p* < 0.01). However, to avoid that domestic cat distribution, strongly linked to human activities, might inflate estimates, the initial dataset was pruned from samples genetically assigned to the domestic cat group and phylogeographic analyses were focused on wildcat and putative admixed populations. Thus, a further MJ network was reconstructed using ND5 sequences from 430 wildcats and 72 putative admixed individuals, corresponding to 21 haplotypes and characterized by 26 polymorphic sites and 21 parsimony informative sites (Figure 2c). In the network, the presence of two main haplogroups, significantly differentiated (ϕ_ST_ = 0.97, *p* < 0.01; Table 2) was still evident: (i) DW, cleaned from “*d*” haplotypes, including 174 wildcats and 61 putative admixed cats; and (ii) W, including 256 wildcats and 11 putative admixed individuals (Figure 2c). 

Haplogroup DW showed a split, roughly corresponding to node 2 of the BT tree (Figure 2c), which separated clades I-II and III. In particular, sub-clade Ia included haplotype *dw*4, which was the most frequent within haplogroup DW, was found in 146 individuals, and showed a spread distribution across central and southern Europe (Figure 3).

Haplogroup W showed a split, roughly corresponding to node 5 of the BT tree (Figure 2c), which separated a clade IV, shared among 85 individuals, most of them collected in *Italy* (63.2%) and the *Iberian peninsula* (18.2%) macro-regions and a clade V, shared among 182 individuals, mostly collected in *Central Europe* (62.6%), and in the *Balkan* (23%) macro-regions (Figure 2c) (see Supplementary Material and Appendix A for details).

The spatial analysis of molecular variance (SAMOVA) showed that the most statistically-supported geographic partition within haplogroup W corresponded to *K* = 4 population groups, with an overall F_CT_ = 0.64 (*p* < 0.01) and 63.2% of variation explained among the detected repartitions. The four groups included Italy and South-Eastern Europe (Cluster 1), Central and North-Eastern Europe, the Balkans (Cluster 2), Scotland (Cluster 3), and the Iberian Peninsula (Cluster 4) (Figure 3).

Conversely, haplogroup DW was optimally structured in *K* = 3 clusters, including Scotland (Cluster 1), the Iberian Peninsula (Cluster 2), and the remaining European populations (Cluster 3) (Figure 3), with a lower overall F_CT_ = 0.28 (*p* < 0.05) and most of the variation explained within populations (63.06%). 

The independent AMOVAs performed considering six biogeographic repartitions ([25]; Figure 1), yielded concordant results but with a slightly lower proportion of variation explained among macro-regions (58.24%) for the haplogroup W and a higher proportion of variance within populations (82.8%) in haplogroup DW (Table 2).

### 3.3. Demographic Analyses

A weak significant sign of population expansion was detected in the domestic cats of lineage D with a near bell-shaped curve in the mismatch plot (Figure 4), a Tajima’s D = −0.413 (*p* = 0.078) and a Fu and Li’s F = −2.337 (*p* < 0.05) (Table 2). The mismatch distribution curve (Figure 4) and the slightly significant negative values of Tajima’s and Fu and Li’s estimators suggested a low degree of population expansion also for lineage W (Table 1). However, within this haplogroup, only *Italy* and the *Balkan* macro-regions presented an increasing trend in the mismatch plot consistent with significant negative values of Tajima and Fu and Li’s statistics (Table 1, Figure 4), although these values were lower than two, suggesting caution in considering a hypothesis of actual expansion [67]. Among *dw* haplotypes, only those of *Central Europe* showed traces of an actual expansion trend with significant negative values of the statistics (Tajima’s D = −2.002; *p* < 0.01 and FU and Li’s F = −1.106; *p* < 0.05; Table 1) and a low peak in the mismatch plot (Figure 4).

### 3.4. Approximate Bayesian Computation Analyses

ABC simulations for haplogroup W provided the best support for scenario 4 (simultaneous population splitting with the following gene flow, Figure 5a) with a posterior probability of 0.372 (95% C.I. 0.000–0.796). Under this scenario, the median values of the divergence time showed that Cluster 1, Cluster 2, and Cluster 3 started isolating about 40,800 generations ago (5–95% quartile: 13,900–53,600). Considering a wildcat generation time of two years [16], the time from the most recent common ancestor (TMRCA) of these populations was estimated at about 81,600 years ago (5–95% quartile: 27,800–107,200 years ago). The following admixture event between Cluster 1 and Cluster 2 contributed to generating Cluster 4 approximately 8920 years ago. Simulations on the DW haplotype (Figure 5b) showed the best posterior probability for scenario 1 (simultaneous splitting in recent times) with a posterior probability of 0.748 (95% C.I. 0.367–1.000). Median values of the divergence time from TMRCA suggested 540 generations ago (5–95% quartile: 42.2–5250), corresponding to about 1080 years ago (5–95% quartile: 84.4–10,500).

## 4. Discussion

Pleistocene climate oscillations significantly shaped the biogeographic patterns and genetic structure of many mammal species within the Palaearctic region [5]. Several examples of shifts in the distribution and genetic composition of mammal populations following glacial and sea-level cycles have been clearly described, highlighting how recurring east–west colonization waves introduced new genetic variants, and subsequent post-glacial recolonizations from Mediterranean and extra-Mediterranean *refugia* further modified their genetic makeup [5,10,68,69,70]. In addition, for some species affected by anthropogenic hybridization, such as the European wildcat (*Felis silvestris*), the wolf (*Canis lupus*), or the wild boar (*Sus scrofa*), the genetic mosaic has been further altered by the introgression of domestic variants [31,71,72]. Therefore, here we describe phylogenetic and phylogeographic patterns obtained by analyzing a diagnostic fragment of the mtDNA on a wide sampling of European wildcats collected across Europe to (i) detect clear signs of genetic differentiation between central and southern European wildcat populations, likely resulting from glacial isolation and consequent post-glacial recolonization processes, and (ii) disentangle the origin of shared haplotypes between wild and domestic cats.

### 4.1. Phylogenetic Patterns 

Although based on partial ND5 sequences, our phylogenetic reconstructions on European wild and domestic cats well reflected their evolutionary relationships confirming previous studies on larger portions of the mitochondrial DNA [20,31,73], as well as entire mitogenomes [74]. Indeed, we detected four main *F. catus*/*lybica* haplogroups (D: Ia, Ib, II, and III), showing an overall high level of haplotype diversity (0.735 ± 0.018) and originating about 80,000 years ago. According to Driscoll et al. [20], such clades might reflect the multiple origins of the *F. catus*/*lybica* lineage, whose estimated coalescence time, although older, is largely included within our confidential intervals. Whitin *F. catus*/*lybica* lineage, clades I and II showed a higher haplotype richness and frequency among samples (in particular, they included 58.2% of all domestic cats). Such haplogroups could be roughly associated with lineages A/B, previously found by Driscoll et al. [20] and Ottoni et al. [31], which represent the Near Eastern *Felis* contribution to the mtDNA pool of present-day domestic cats. On the other hand, clade III, showing a basal phyletic position, might correspond to lineage C described by Driscoll et al. [20] and Ottoni et al. [31], which includes north African wildcat haplotypes (possibly of ancient Egypt derivation) later spread in Europe [31].

Concordant results with previous studies [20,31,73] were further found for the *F. silvestris* lineage W, which showed two main clades, IV and V, dating back about 62,400 years ago. Similar divergent times and phylogenetic patterns have been observed also in other species, such as the pine marten (*Martes martes*) [69] and the wolf (*Canis lupus*) [75] and might be the result of Pleistocene climatic oscillations on mammal population distribution and evolution [5,69,75,76].

### 4.2. Phylogeographic Patterns

A spatial analysis of molecular variance (SAMOVA) was performed to clarify phylogeographic patterns within the European wildcat lineage W, showing that its haplotypes were spatially clustered into at least four main geographic groups roughly concordant with the biogeographic regions previously detected analyzing nuclear markers [25,38]: (1) a South-Eastern European cluster, spanning from Italy to Hungary; (2) a Central European cluster, spanning from the Italian and Balkan Alps to Germany; (3) a cluster including the isolated Scottish population; 4) a cluster including the Iberian population. In particular, Cluster 1 showed the overall lowest genetic variability. However, further analyses, including additional wildcat samples from other unsampled and formerly in contact with eastern Countries, such as Anatolia, might reveal increased genetic variability levels and shed light on the possible contributions of such populations in shaping the current European wildcat evolutionary history in eastern European populations, such as that living in Hungary. This cluster was mainly represented in clade IV with (a) the dominant haplotype *w1*, shared at low frequencies also with the Iberian Peninsula wildcats, which could be the result of a colonization wave from eastern Europe during glacial periods and subsequent isolation south of the Alps, which acted as a natural barrier to gene flow [23,25,77], and (b) the presence of a private haplotype *w5* in Sicily, that could result from the long-lasting isolation of the Island from the Peninsula [78]. These recolonization and subsequent isolation patterns are further confirmed by the presence of shared mtDNA haplotypes in other mammal species, such as the pine marten (*Martes martes*) [69], the red deer (*Cervus elaphus*) [79], the roe deer (*Capreolus capreolus*) [5], and the brown bear (*Ursus arctos*) [80] in southern Europe.

Cluster 2 showed a higher genetic variability, and was mainly represented in clade V, characterized by the predominance of haplotype *w4*, which might have originated in extra-Mediterranean or in the Dinaric–Alpine *refugia* and successively widespread in Central Europe, as hypothesized through molecular studies carried out also on the hedgehog (*Erinaceus europaeus*) [81], the Eurasian lynx (*Lynx lynx*) [82] and the red deer *(Cervus elaphus)* [9] and confirmed by the absence of Pleniglacial wildcat archaeological findings [23].

Cluster 3 showed a single unique haplotype *w3*, which might be the legacy of the mtDNA gene pool of a continental wild ancestor which migrated to Britain via land bridge approximately 10,000 years ago [83] and originated the Island population, which successively experienced recurrent bottlenecks during glacial maximums, more recent demographic declines, and a compromising anthropogenic admixture with domestic cats [84]. Finally, Cluster 4 showed a) the balanced presence of haplogroups IV and V, which might derive from the Pleistocene population migration waves from Central Europe, as supported also by ABC analyses, and b) the occurrence of three private haplotypes, *w9* and *w12* in Spain and *w8* in Portugal, which might have originated during the subsequent isolation south of the Pyrenes [85,86]. 

### 4.3. Mito-Nuclear Discordance and Evolutionary Hypotheses

Our phylogenetic and phylogeographic history of the European wildcat assessed by partial mitochondrial sequences revealed the widespread presence of haplotypes shared between wild and domestic cat populations. Indeed, a consistent proportion of individuals previously assigned to the wildcat population through STR Bayesian clustering analyses [25] were included in the mitochondrial lineage D. Cases of *Felis* mito-nuclear discordances have been already described and generally attributed to recent *F. catus* mitochondrial introgressions, which may have likely eroded domestic ancestry at the nuclear *loci* after a few backcrossing generations [87], leaving exogenous traces only at the mtDNA or at a small portion of the nuclear genome [16]. Accordingly, our mtDNA data showed several shared haplotypes (*dw1, dw2, dw3, dw5, dw7*), mostly frequent within domestic cats, which seem to have recently differentiated and simultaneously split about 1000 years ago, as further revealed by the best ABC evolutionary scenario, supporting the hypothesis of an *F. catus* introgression. 

However, other two shared haplotypes, belonging to the D subclade Ia (*dw4* and *dw6*), mostly frequent within wildcats, seem to have originated about 37,000 years ago, which, according to the last available archaeological and genetic findings, long pre-dated any evidence for cat domestication [20,30,88], thus suggesting their possible natural *F. lybica* derivation. This latter hypothesis fits well with the scenario of a late Pleistocene European wildcat migration toward the Levant and Anatolia regions already occupied by *F. lybica* populations [89,90]. Such a syntopic event might have promoted a natural inter-*taxon* gene flow introgressing *F. lybica* mitochondrial signatures in some European wildcat individuals [31]. 

A possible female-biased directionality of the admixture patterns might justify the high numbers of *F. catus*/*lybica* mtDNA hyplotypes found in the European wildcats analyzed in this study, though this hypothesis should be confirmed or denied by further Y-chromosome haplotype analyses.

## 5. Conclusions

In this study, using a short but highly diagnostic portion of the mtDNA, we provide the first exhaustive description of the European wildcat phylogenetic and phylogeographic structure across the entire species’ range in the continent. Our results suggest the presence of at least three main continental biogeographic clusters, roughly corresponding to the Iberian Peninsula, the South-Eastern European Mainland, and Central Europe, whose origin fits well with a model of species glacial isolation and post-glacial re-expansion from the Mediterranean and extra-Mediterranean *refugia* during the late Pleistocene. As expected, a fourth biogeographic cluster was identified in the isolated and almost genetically compromised Scotland wildcat population [84], which showed unique wild and domestic haplotypes. Based on their wild ancestry content, such biogeographic clusters could be used to identify four possible preliminary corresponding Conservation Units (CU, [91]) to be treated as different management priorities and preserved through well-planned conservation actions, depending on their wild genomic mito-nuclear concordance. Interestingly, our data also show the presence of mtDNA haplotypes shared between wild and domestic cat populations, likely resulting from two different independent evolutionary processes, historical natural gene flow among wild lineages, and recent wild *x* domestic anthropogenic hybridization. Future studies, based on the analyses of entire mitogenomes and whole nuclear genomes of early domesticated cats from museum collections, modern and ancient European and African wildcats collected also in their overlapping distribution ranges (from Turkey to the Near East), could undoubtedly help researchers to disentangle this complex biogeographic mosaic, clarify the evolutionary histories and admixture patterns, as well as shed light on the origin of the current mito-nuclear variability of the European wildcat populations and their long-term adaptive potential.

## Figures and Tables

**Figure 1 animals-13-00953-f001:**
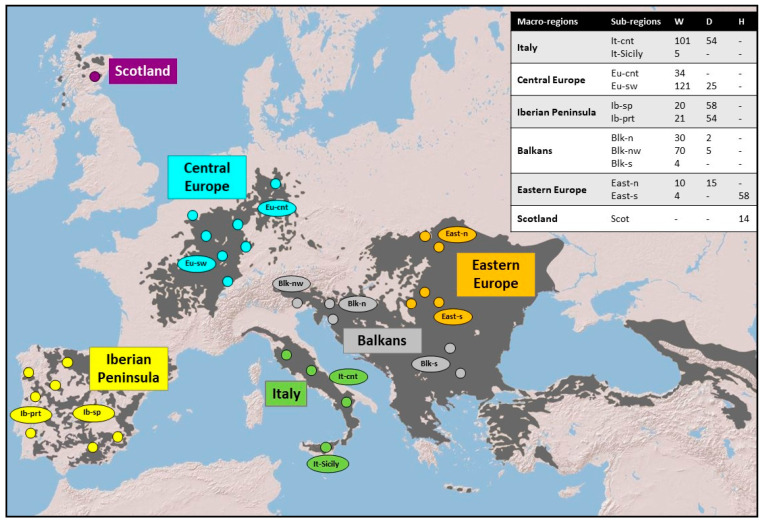
Map of the approximate cat sampling locations. Dark grey areas correspond to the most recently updated IUCN *Felis silvestris* distribution range [46]. Macro-regions (Italy, Central Europe, Iberian Peninsula, Balkans, Eastern Europe, Scotland) are indicated in different colors. Each macro-region was further subdivided into sub-regions (It-cnt: Central Italy; It-Sicily: Sicily; Eu-cnt: Center–North Germany; Eu-sw: Southern and Western regions of Central Europe; Ib-prt: Portugal; Ib-sp: Spain; Balk-n: Northern Balkans; Balk-s: Southern Balkans; Balk-nw Eastern Italian Alps and North-Western Dinarides; East-n: Poland; East-s: Hungary). Samples from the North-Eastern Italian Alps were included in the Balkans group (blk-nw) according to the most recent genetic structure detected in Europe through Bayesian assignment procedures performed using nuclear microsatellite genetic profiles [25]. Small circles indicate the approximate geographic locations within the main sampling areas. Each sub-region is provided with the number of analyzed samples classified on the base of their 31-STR assignment values [25]: W = wildcat; D = domestic cat; H = putative admixed. Further details are available in Appendix A.

**Figure 2 animals-13-00953-f002:**
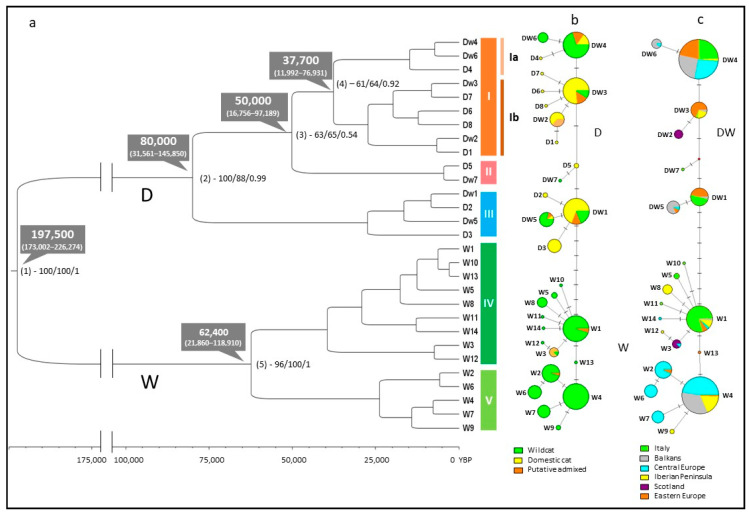
(**a**) Bayesian mtDNA tree (BT) computed in Beast [57] showing the phylogenetic relationships among ND5 haplotypes obtained from all the 715 analyzed cat samples. Capitalized letters (D, W) indicate the two main lineages. Roman numerals (I–V) in coloured box indicate the main clades. Grey cartoons indicate the estimated node divergence times, whose 95% posterior density confidential intervals are reported in brackets. Annotated nodes show the Bayesian highest posterior density values and also the respective bootstrap values derived from the NJ and ML trees. (**b**) MJ networks among ND5 haplotypes were obtained from all 715 analyzed cat samples. Haplotypes, according to the 31-STR Bayesian assignment values of their belonging samples [25], were partitioned and colored as domestic, wild, and putative admixed groups. (**c**) MJ networks among ND5 haplotypes, including only wildcats and putative admixed individuals (n = 502). Each haplotype was divided and colored according to the proportion of individuals belonging to each biogeographic macro-region. Small bars indicate the number of mutations (greater than one) between two different haplotypes. The frequency of each haplotype is proportional to the size of the circles.

**Figure 3 animals-13-00953-f003:**
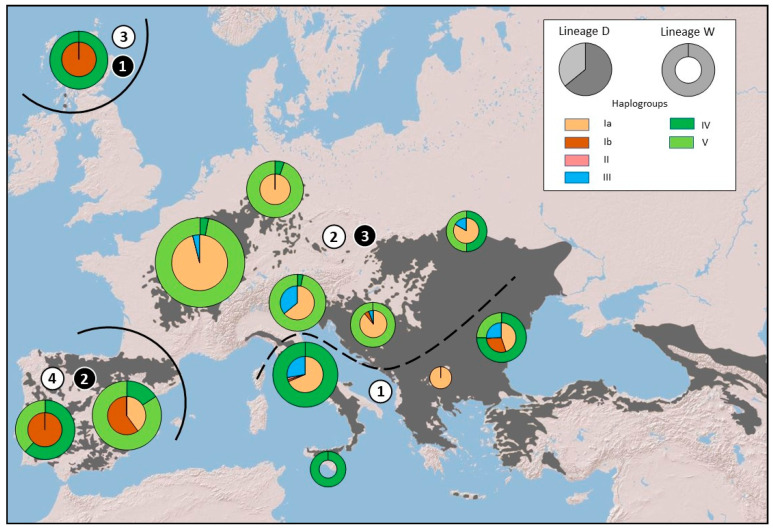
Distribution map of the main six haplogroup clades derived from the Bayesian tree. Donut charts show the proportional frequency of haplogroups of lineage W. The inscribed pie charts show the proportional frequency of lineage DW (lineage D pruned by domestic cats). Circles are approximately proportional to sample size. Black solid curved lines trace the geographical separations among genetic clusters detected in both lineages W and DW through SAMOVA analyses, while dotted line shows the additional genetic separation found for lineage W. Black numbers in white circles indicate the clusters found by SAMOVA analyses for lineage W. White numbers in black circles show the clusters found by SAMOVA analyses for lineage DW.

**Figure 4 animals-13-00953-f004:**
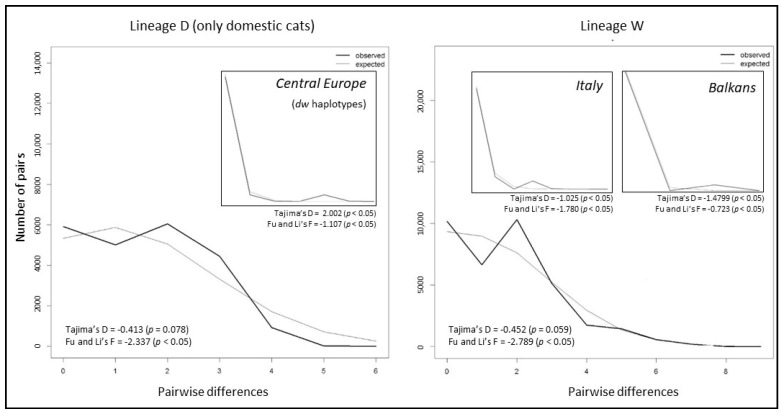
Mismatch distribution plots computed using haplogroups showing significant values of Tajima’s D and Fu and Li’s F indexes. Larger plots show mismatch distribution results about lineage D (considering only domestic cat samples) and lineage W. Smaller plots show mismatch distribution results about different macro-regions within each of the two lineages.

**Figure 5 animals-13-00953-f005:**
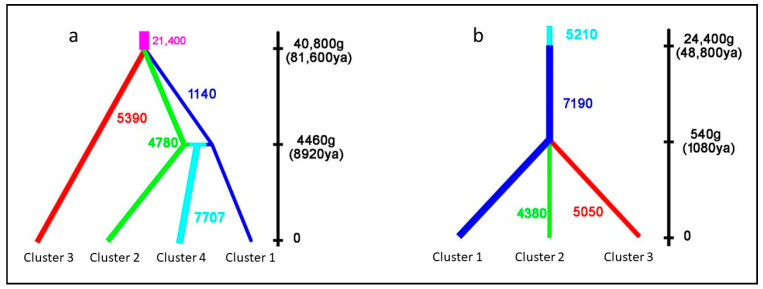
Graphical representation of the resulting population sizes and divergence times estimated for the two best-simulated scenarios inferred by ABC simulations, using a generation time g = 2 years. The width of branches is proportional to the inferred effective population sizes. (**a**) The best scenario was inferred using lineage W haplotypes. (**b**) The best scenario was inferred using lineage DW (lineage D pruned by domestic cats) haplotypes. Cluster numbers refer to the best *K* repartition obtained by SAMOVA analyses for each of the two lineages.

**Table 1 animals-13-00953-t001:** Summary of genetic variability statistics obtained by analyzing a portion of the mitochondrial ND5 gene and considering as sample groups: (a) STR Bayesian assignment cat populations [25]; (b) main phylogenetic lineages; and (c) biogeographic macro-regions within each phylogenetic lineage. For lineages W and DW (lineage D pruned by domestic cats), the statistics are also provided for each macro-region. N = number of samples; SD = standard deviation.

	N	Number of Haplotypes	Nucleotide Diversity (π ± SD)			Haplotype Diversity (h ± SD)			Tajima’s D	*p*-Value	Fu and Li’s F	*p*-Value
**Overall**	**715**	**28**	**0.870**	**±**	**0.013**	**0.862**	**±**	**0.006**				
Domestic cats	213	12	0.230	±	0.150	0.735	±	0.018	−0.413	0.078	−2.337	0.026
Putative admixed	72	8	0.686	±	0.378	0.797	±	0.027	1.145	0.173	4.129	0.442
Wildcats	430	20	1.063	±	0.552	0.813	±	0.011	2.297	0.405	4.534	0.280
**Lineage D**	**448**	**14**	**0.247**	**±**	**0.161**	**0.746**	**±**	**0.012**	**−0.347**	**0.067**	**−2.271**	**0.046**
**Lineage W**	**267**	**14**	**0.255**	**±**	**0.166**	**0.720**	**±**	**0.021**	**−0.452**	**0.059**	**−2.789**	**0.030**
Italy	55	4	0.048	±	0.055	0.173	±	0.068	−1.025	0.043	−1.780	0.009
Central Europe	117	7	0.165	±	0.121	0.676	±	0.034	−0.359	0.098	−0.706	0.103
Iberian Peninsula	40	5	0.245	±	0.164	0.637	±	0.063	0.434	0.135	1.091	0.260
Balkans	43	2	0.014	±	0.028	0.047	±	0.044	−1.480	0.001	−0.723	0.042
Eastern Europe	8	4	0.230	±	0.174	0.750	±	0.139	1.347	0.234	−0.375	0.030
Scotland	7	1										
**Lineage DW**	**235**	**7**	**0.208**	**±**	**0.142**	**0.581**	**±**	**0.034**	**0.415**	**0.223**	**0.742**	**0.202**
Italy	54	4	0.207	±	0.144	0.461	±	0.058	1.288	0.359	1.942	0.388
Central Europe	41	3	0.037	±	0.047	0.096	±	0.062	−2.002	0.000	−1.107	0.047
Iberian Peninsula	8	2	0.064	±	0.073	0.429	±	0.169	0.334	0.479	0.536	0.132
Balkans	61	5	0.260	±	0.170	0.571	±	0.055	1.461	0.366	1.828	0.341
Eastern Europe	64	4	0.199	±	0.139	0.656	±	0.035	1.235	0.384	1.993	0.406
Scotland	7	1										

**Table 2 animals-13-00953-t002:** Analysis of molecular variance (AMOVA) on a portion of the mitochondrial ND5 gene computed among and within three different sample groups: (a) the STR Bayesian assignment populations [25]; (b) the main phylogenetic lineages; and (c) the biogeographic macro-regions within each phylogenetic lineage. ϕ_ST_: differences among groups; ϕ_SC_: differences among populations within a group. All values were highly significant (*p* < 0.05). Lineage DW represents lineage D pruned by domestic cats.

Source of Variation	Variance Components	Percentage of Variation	Differentiation Indexes
Among wildcats, domestic cats, and putative admixed	1.51	36.66	ϕ_ST_ = 0.37
Within wildcats, domestic cats, and putative admixed	2.60	63.34	
Among lineages W/DW	5.68	86.68	ϕ_ST_ = 0.92
Among macro-regions/within lineages	0.37	5.60	ϕ_SC_ = 0.42
Within macro-regions	0.51	7.72	
Lineage W			
Among macro-regions	0.59	58.24	ϕ_ST_ = 0.58
Within macro-regions	0.42	41.76	
Lineage DW			
Among macro-regions	0.12	17.21	ϕ_ST_ = 0.17
Within macro-regions	0.60	82.79	

## Data Availability

Data presented in this study are available on request.

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
