# Peer review of "Phylogenetic History and Phylogeographic Patterns of the European Wildcat (Felis silvestris) Populations"

_animals, 2023, doi:10.3390/ani13050953_

Round 1
Reviewer 1 Report
An excellent and fully necessary research for a good conservation in practice of the European wildcats, particularly since it highlights the importance of the decreasing and threatened Mediterranean populations of the Iberian Peninsula. I only have a minor question related to sample distribution: While I can understand the case of Scotland, why did authors use only admixed cats in Eastern Europe-S? I suppose that it is due to the absence of wildcat samples, but this evident sampling bias should be argued, particularly because Eastern Europe-S should be a key glacial refugia. In consequence, a higher diversity of the W lineage in Eastern Europe should be expected, particularly if this population was formerly in contact with the Anatolian region. So, I wonder if this low genetic variability found by this study could be related with the observed sampling bias.
Author Response
Response to Reviewer 1 Comments
Dear Reviewer,
Thank you very much for evaluating our manuscript titled “Phylogenetic History and Phylogeographic Patterns of the European Wildcat (Felis silvestris) Populations”. Please find enclosed our answers to your comments and considerations.
Question - An excellent and fully necessary research for a good conservation in practice of the European wildcats, particularly since it highlights the importance of the decreasing and threatened Mediterranean populations of the Iberian Peninsula.
Reply - Thank you very much for appreciating our work and its potential conservation management value, especially for isolated populations such as the one living in the Iberian Peninsula.
Question - I only have a minor question related to sample distribution: While I can understand the case of Scotland, why did authors use only admixed cats in Eastern Europe-S? I suppose that it is due to the absence of wildcat samples, but this evident sampling bias should be argued, particularly because Eastern Europe-S should be a key glacial refugia.
Reply - Thank you for this very useful comment. We used all the available samples collected over years in Hungary and most of them, as already shown in the scientific literature (Pierpaoli et al. 2003; Mattucci et al. 2016), were classified as putative admixed individuals through Bayesian assignment procedures with microsatellite genotypes, confirming the high levels of domestic introgression in the Hungarian wildcat population. We added some details in the Materials and Methods section to justify the high numbers of admixed individuals analysed from Hungary and Scotland sub-regions. We further added some cautions in the Discussion section about the possibility of unsampled populations from Eastern Europe which might better explain the origin of Hungary and East-European populations.
Question - In consequence, a higher diversity of the W lineage in Eastern Europe should be expected, particularly if this population was formerly in contact with the Anatolian region. So, I wonder if this low genetic variability found by this study could be related with the observed sampling bias.
Reply – Thank you very much for this note. An overall low genetic variability was detected when considering the entire Cluster 1, which includes Italy and South-East Europe populations. We agree with the referee’s comment since the single Eastern Europe macro-region population, which includes Hungary and Poland populations, showed both high Haplotype diversity (0.750±0.139) and high Nucleotide diversity (0.230±0.174) values. Therefore, an eventual inclusion of wildcat samples from other Eastern Countries, including Anatolia with which the Eastern European population was formerly in contact, might show increasing Eastern Europe macro-region genetic variability levels and shed light on the possible contributions of populations from such regions in shaping the current European wildcat evolutionary history. We emphasised this points in the Discussion and Conclusions sections.
References
Mattucci, F.; Oliveira, R.; Lyons, L.A.; Alves, P.C.; Randi, E. European Wildcat Populations Are Subdivided into Five Main Biogeographic Groups: Consequences of Pleistocene Climate Changes or Recent Anthropogenic Fragmentation? Ecol. Evol. 2016, 6, 3–22, doi:10.1002/ece3.1815
Pierpaoli, M.; Biro, Z.S.; Herrmann, M.; Hupe, K.; Fernandes, M.; Ragni, B.; Szemethy, L.; Randi, E. Genetic Distinction of Wildcat (Felis silvestris) Populations in Europe, and Hybridization with Domestic Cats in Hungary. Mol. Ecol. 2003, 12, 2585–2598, doi:10.1046/j.1365-294X.2003.01939.x

Reviewer 2 Report
The manuscript by Velli et al. entitled “Phylogenetic History and Phylogeographic Patterns of the European Wildcat (Felis silvestris) Populations” is aiming to describe phylogeographic patterns of mtDNA in wild cats and complement the knowledge of hybridization between wild and domestic cats.The authors sequenced hundreds of cats and used the nuclear information from the previous publication of their team. Overall, it is a very interesting study with robust and detailed phylogenetic analyses and is worthy of publishing. My main concerns are the readability of the text and DIY analysis.
In some parts, the text suffers from the overwhelming detailed description of results. One page is given to the description of the phylogenetic tree and the distribution of haplotypes. Most of the information is visible from the figure and the main message is lost. The same applies to networks. Try to adjust the text, provide it with more readability, and highlight the most important outcomes, all the other results can be just referenced to figures or supplementary materials.
It's a shame that the authors didn't use both mitochondrial and STR data if those were available. Many analyses can use both of them and the comparison of the results might be truly interesting. For example, DIY could highly benefit from comparing nuclear and mitochondrial data or better use them together.
Authors are not really testing the population differentiation if they only use a priory defined populations. Did the authors try to use some landscape genetic approaches (like geneland, alleles in space)? This could also benefit from using both mt and STR data.
I am not sure I fully follow the decision-making process of creating the scenarios for DIY. Can the authors explain in more detail why those scenarios were selected and some thers omitted?
Minor comments:
The first figure would benefit from adding sampling localities. I can see it was included in the first paper in 2016 so adjusting the map should be no problem.
Figure two should contain the confidence intervals of the timing.
Author Response
Response to Reviewer 2 Comments
Dear Reviewer,
Thank you very much for evaluating our manuscript titled “Phylogenetic History and Phylogeographic Patterns of the European Wildcat (Felis silvestris) Populations”. Please find enclosed our answers to your comments and considerations.
Question - The manuscript by Velli et al. entitled “Phylogenetic History and Phylogeographic Patterns of the European Wildcat (Felis silvestris) Populations” is aiming to describe phylogeographic patterns of mtDNA in wild cats and complement the knowledge of hybridization between wild and domestic cats. The authors sequenced hundreds of cats and used the nuclear information from the previous publication of their team. Overall, it is a very interesting study with robust and detailed phylogenetic analyses and is worthy of publishing. My main concerns are the readability of the text and DIY analysis.
Reply - Thank you very much for revising our work and providing positive and constructive comments. We tried to improve the readability of the text, especially in the Results sections. Furthermore, we better clarified the choice of the DIY prior scenarios used in performing the Approximate Bayesian Computation (ABC) simulations and the reasons why using only a high diagnostic portion of the mitochondrial DNA to investigate wildcat population demographic scenarios and estimate divergence times.
Question - In some parts, the text suffers from the overwhelming detailed description of results. One page is given to the description of the phylogenetic tree and the distribution of haplotypes. Most of the information is visible from the figure and the main message is lost. The same applies to networks. Try to adjust the text, provide it with more readability, and highlight the most important outcomes, all the other results can be just referenced to figures or supplementary materials.
Reply - Thank you. We agree, so we tried to improve the readability of the text, better summarising the description of the Bayesian phylogenetic tree and the distribution of haplotypes, retaining the most important information. We added the details in the Supplementary Material section.
Question - It's a shame that the authors didn't use both mitochondrial and STR data if those were available. Many analyses can use both of them and the comparison of the results might be truly interesting. For example, DIY could highly benefit from comparing nuclear and mitochondrial data or better use them together.
Reply – We thank the referee for this note since we think that the combined use of mitochondrial and STR data would provide useful results. However, we preferred to focus our study on the sole mtDNA variability since, for our knowledge, a phylogenetic and phylogeographic reconstruction study based on the analysis of such marker at the Continental scale still lacks. Our results might be useful to complement the European phylogeographic findings based on STR data carried out by Mattucci et al. (2016) with which they revealed to be fully concordant. In this way, we are confident enough that both studies will permit to share a common overview while maintaining their own readability and level of details. Nonetheless, as we highlighted in the Conclusion section, we believe that future studies, including samples from other unsampled and formerly in contact Eastern Countries, such as Anatolia and middle-east regions, and combining more powerful genetic markers such as large panels of SNPs (Mattucci et al. 2019), entire mitogenomes or even complete nuclear genomes could give additional and more detailed pivotal information to clarify the current genetic mosaic of the European wildcat and design appropriate management and conservation actions.
Question - Authors are not really testing the population differentiation if they only use a priory defined populations. Did the authors try to use some landscape genetic approaches (like GeneLand, alleles in space)? This could also benefit from using both mt and STR data.
Reply - We agree about the potential benefits in using a landscape genetic approach. However, the heterogeneous nature of our sampling (methodologically and temporally) and the approximate geographic location of many of our collected materials make difficult the use of landscape approaches such as the one implemented in Geneland, with a hight risk of unreliable results. We investigated mtDNA population differentiation using the macro-regions previously detected by Mattucci et al. (2016) through STR Bayesian clustering analyses performed using the same sample dataset. Additionally, such geographic groups were further taken into account to perform downstream spatial analyses of molecular variance (SAMOVA) which defined the best number of K clusters of mtDNA geographically homogeneous populations.
Question - I am not sure I fully follow the decision-making process of creating the scenarios for DIY. Can the authors explain in more detail why those scenarios were selected and some others omitted?
Reply – Thank you for this point. As suggested, we better clarified in the Materials and Methods section our criteria to select tested populations and design evolutionary scenarios. Indeed, we better explained that, to avoid over-computation efforts, we tried to design the smallest number of plausible evolutionary scenarios. Therefore, we used as prior populations the mtDNA clusters found through SAMOVA analyses. We designed alternative evolutionary hypotheses using haplotype distribution and divergence times estimated from the Bayesian tree phylogeny and modelled population dynamics taking into account a) the main phylogeographic hypotheses reported in the most updated scientific literature (e.g: Mattucci et al. 2016; Mousavi et al. 2022) and b) the possible recent role of anthropogenic hybridization as a main driver for the spread of domestic haplotypes of the lineage DW into the wildcat population (Driscoll et al. 2007; Ottoni et al. 2017).
Minor comments:
Question - The first figure would benefit from adding sampling localities. I can see it was included in the first paper in 2016 so adjusting the map should be no problem.
Reply - Thank you, we modified the image as suggested.
Question - Figure two should contain the confidence intervals of the timing.
Reply - Thank you, we modified the image as suggested.
References
Driscoll, C.A.; Menotti-Raymond, M.; Roca, A.L.; Hupe, K.; Johnson, W.E.; Geffen, E.; Harley, E.H.; Delibes, M.; Pontier, D.; Kitchener, A.C.; et al. The Near Eastern Origin of Cat Domestication. Science. 2007, 317, 519–523, doi:10.1126/science.1139518.
Mattucci, F.; Galaverni, M.; Lyons, L.A.; Alves, P.C.; Randi, E.; Velli, E.; Pagani, L.; Caniglia, R. Genomic Approaches to Identify Hybrids and Estimate Admixture Times in European Wildcat Populations. Sci. Rep. 2019, 9, 1–15, doi:10.1038/s41598-019-48002-w.
Mattucci, F.; Oliveira, R.; Lyons, L.A.; Alves, P.C.; Randi, E. European Wildcat Populations Are Subdivided into Five Main Biogeographic Groups: Consequences of Pleistocene Climate Changes or Recent Anthropogenic Fragmentation? Ecol. Evol. 2016, 6, 3–22, doi:10.1002/ece3.1815
Mousavi, M.; Naderi, S.; Rezaei, H.R.; Adibi, M.A. Evolutionary History and Distribution of African Wildcat, Felis lybica in Iran. Casp. J. Environ. Sci. 2022, 20, 637–648, doi:10.22124/cjes.2022.5708
Ottoni, C.; Van Neer, W.; De Cupere, B.; Daligault, J.; Guimaraes, S.; Peters, J.; Spassov, N.; Prendergast, M.E.; Boivin, N.; Morales-Muñiz, A.; et al. The Palaeogenetics of Cat Dispersal in the Ancient World. Nat. Ecol. Evol. 2017, 1, 0139, doi:10.1038/s41559-017-0139.

Round 2
Reviewer 2 Report
The manuscript was revised by the authors and the majority of my concerns were adequately solved. A combination of nuclear and mitochondrial data was not accepted, but it doesnt decrease the value of the findings. I have no further comments and I congratulate to authors.